# The Role of the miR-17-92 Cluster in Autophagy and Atherosclerosis Supports Its Link to Lysosomal Storage Diseases

**DOI:** 10.3390/cells11192991

**Published:** 2022-09-26

**Authors:** Daniel Ortuño-Sahagún, Julia Enterría-Rosales, Vanesa Izquierdo, Christian Griñán-Ferré, Mercè Pallàs, Celia González-Castillo

**Affiliations:** 1Laboratorio de Neuroinmunobiología Molecular, Instituto de Investigación en Ciencias Biomédicas (IICB) CUCS, Universidad de Guadalajara, Guadalajara 44340, Jalisco, Mexico; 2Tecnologico de Monterrey, Escuela de Medicina y Ciencias de la Salud, Campus Guadalajara, Zapopan 45201, Jalisco, Mexico; 3Pharmacology and Toxicology Section and Institute of Neuroscience, Faculty of Pharmacy and Food Sciences, University of Barcelona, 08007 Barcelona, Spain

**Keywords:** autophagy, cholesterol, enzyme deficiency, lysosomal storage diseases, metabolism, vesicle trafficking

## Abstract

Establishing the role of non-coding RNA (ncRNA), especially microRNAs (miRNAs), in the regulation of cell function constitutes a current research challenge. Two to six miRNAs can act in clusters; particularly, the miR-17-92 family, composed of miR-17, miR-18a, miR-19a, miR-20a, miR-19b-1, and miR-92a is well-characterized. This cluster functions during embryonic development in cell differentiation, growth, development, and morphogenesis and is an established oncogenic cluster. However, its role in the regulation of cellular metabolism, mainly in lipid metabolism and autophagy, has received less attention. Here, we argue that the miR-17-92 cluster is highly relevant for these two processes, and thus, could be involved in the study of pathologies derived from lysosomal deficiencies. Lysosomes are related to both processes, as they control cholesterol flux and regulate autophagy. Accordingly, we compiled, analyzed, and discussed current evidence that highlights the cluster’s fundamental role in regulating cellular energetic metabolism (mainly lipid and cholesterol flux) and atherosclerosis, as well as its critical participation in autophagy regulation. Because these processes are closely related to lysosomes, we also provide experimental data from the literature to support our proposal that the miR-17-92 cluster could be involved in the pathogenesis and effects of lysosomal storage diseases (LSD).

## 1. Introduction

Establishing the role of non-coding RNA (ncRNA) in the regulation of cell function has been one of the new research challenges of the 21st century, with microRNAs, commonly abbreviated as miRNAs, constituting a target of great interest. The miRNAs are small ncRNA fragments (about 22 nucleotides in length) that regulate gene expression at the post-transcriptional level. They are known to modify gene function by binding to specific fragments of messenger RNA (mRNA) and suppressing their translation or promoting their degradation to ultimately repress gene expression [1]. To date, about 38,589 miRNAs have been introduced into the miRBase [2].

Although it has been described that most miRNAs can act by themselves, some of them act in clusters, composed of at least two miRNAs and up to six, which share a proximal location in the genomic DNA [3]. This is the case for the miR-17-92 cluster. It is composed of six miRNAs; miR-17, miR-18a, miR-19a, miR-20a, miR-19b-1, and miR-92a (or miR-92-1) and is located on chromosome 13 in humans and on chromosome 14 in mice. This cluster of six miRNAs regulates certain functions during embryonic development, including cell differentiation, growth and development, and morphogenesis [4], both as a cluster and individually. Members of miR-17 family, which share the seed sequence AAACUG, are located on three different clusters: miR-17-92 (six members), miR-106b-25 (three members: miR-106b, miR-93, and miR-25), and miR-106a-363 (six members: miR-106a, miR-18b, miR-20b, miR-19b2, miR-92a-2, and miR-363). These clusters contain members of another three miRNA families: miR-18 family (seed sequence AAGGUG), mir-19 family (seed sequence GUGCAA), and mir-92 family (seed sequence AUUCGA) [5].

## 2. Role of miR-17-92 Cluster Members in Regulating Cellular Energetic Metabolism

Plentiful works have reported that the miR-17-92 cluster, also known as oncomiR-1, and its members, are overexpressed in several types of cancers and constitute, to date, one of the best-characterized oncogenic miRNAs clusters [6]. In addition, it has been identified as a crucial factor involved in the regulation of vascular integrity and angiogenesis [7,8], which mainly regulates endothelial cell function under pathophysiological conditions [9]. Nevertheless, the targetome of miR-17-92 is large and variated [10]. 

Interestingly, this cluster of miRNAs has been implicated as a key factor in the regulation of metabolic reprogramming of tumors. When its expression is absent, cell metabolism decreases, and when it is overexpressed, nutrient utilization by tumor cells increases [11], thus regulating glycolytic and mitochondrial metabolism. Therefore, the members of this cluster were also recently considered mitomiRs, or miRNAs present in mitochondria [12]. Additionally, the miR-17-92 cluster has been recognized as a specific biomarker in gestational diabetes mellitus (GDM) [13,14,15], which increases the interest in the possible participation of this cluster in regulating cellular energetic metabolism.

The regulation of lipid metabolism constitutes a relevant part of metabolism and homeostasis, not only for energy production but also for lipid neogenesis and cholesterol metabolism. Interestingly, a few studies have proposed a relationship between the miR-17-92 family and lipid metabolism regulation. This family has been identified as actively participating in dyslipidemia pathophysiology in patients with coronary artery disease (CAD) [16], as well as in the process of steatosis in hepatic cells, specifically through miR-17, which regulates the expression of *CYP7A1* (cytochrome P450 family 7 subfamily A member 1) and is a regulator of hepatic lipid metabolism [17]. However, the findings available so far only suggest a potential role of miR-17-92 cluster in the regulation of cellular lipid metabolism. Although individual members of the miR-17-92 family modulates lipid metabolism, whether the miR-17-92 cluster functions as a whole remains largely unexplored. 

Moreover, it has been described that dysregulation in the expression of lysosome-related genes may contribute to the incapacity of lysosomes to deal with high lipid consumption and the later development of atherosclerosis, which is caused by cholesterol accumulation [18]. In this case, miRNAs regulate atherosclerotic disease initiation and progression [19], which makes them suitable candidates as biomarkers for atherosclerosis as well as targets for pharmacological intervention as a hopeful therapeutic notion.

Therefore, an alteration in the regulation of lipid metabolism can lead to dysfunction of the lysosomes and, subsequently, generation of atherosclerosis. Given that in addition to controlling cholesterol efflux, lysosomes also regulate autophagy, it is logical to suppose that a possible link between these two cellular processes exists. This link may be represented, at least partially, by the coordinating regulatory action of miR-17-92 family members on gene expression and cell function. Experimental evidence from the literature supporting this hypothesis is presented below, first for atherosclerosis and then for autophagy. 

### 2.1. MiR-17-92 Family Members Have Been Identified as Key Regulators in Atherosclerosis

The plasma expression level of miR-17-5p is increased in patients with CAD, while, interestingly, *VLDLR* (very low-density lipoprotein receptor) mRNA expression is decreased in peripheral blood lymphocytes in these patients [20]. 

Another target repressed by miR-17-5p is *ABCA1* (adenosine triphosphate (ATP)-binding cassette subfamily A member 1) [21], whose expression is the rate-limiting step in reverse cholesterol transport [22], which is considered the only mechanism by which the human body can clear excess cholesterol [23]. On the other hand, miR-17-5p can be downregulated by a hypoxia-induced long non-coding RNA (lncRNA) named *MALAT1* (metastasis-associated lung adenocarcinoma transcript 1) to regulate the expression level of *ABCA1* and then reduce cholesterol accumulation in oxidized-LDL (low-density lipoprotein)-induced macrophages. Therefore, the knockdown of *MALAT1* may promote cholesterol accumulation through the miR-17-5p/*ABCA1* axis [24]. Similarly, the knockdown of miR-17-5p can attenuate atherosclerosis [25] by suppressing inflammation and reducing lipid accumulation in atherosclerotic lesions [21]. Likewise, downregulation of miR-17-5p could be conducted by p53-dependent lincRNA-p21 expression, which consequently protects against atherosclerosis progression via *SIRT7* (sirtuin 7) elevation, which is one of miR-17-5p’s targets [26]. 

Another member of the miR-17-92 cluster, miR-20a/b, together with other miRNAs, has been described as a regulator of *ABCA1* expression: miR-20a/b decreases *ABCA1* expression, thus increasing cholesterol accumulation in vitro. By inhibiting miR-20a/b, *ABCA1* expression and cholesterol flux increase [27]; miR-20a also targets the *PTEN* (phosphatase and tensin homolog) gene, participating in the prevention of CAD by promoting the survival and proliferation of vein endothelial cells. Likewise, overexpression of miR-20a could down- or upregulate the expression levels of several atherosclerosis-related genes in CAD [28]. Furthermore, miR-20a was predicted to regulate myogenic factor 5 (*MYF5*) and proliferator-activated receptor-γ (*PPARγ*), two of the most important genes involved in adipogenesis in both mice and humans [29]. Another target of miR-20a is *LDLR* (low-density lipoprotein receptor) [30], involved in cholesterol homeostasis. Additionally, miR-20a-5p, which targets almost 500 genes [31], has been involved in non-alcoholic fatty liver disease (NAFLD) and in its advanced version, non-alcoholic steatohepatitis (NASH), which is characterized by steatosis, inflammation, ballooning, and fibrosis [32]. Downregulation of miR-20a-5p is accompanied by upregulation of one of its targets, *CD36* (CD36 molecule), and increased lipid deposition, suggesting novel pathogenesis of non-alcoholic fatty liver disease and a potential therapeutic strategy for metabolic diseases [33]. 

A third member of the cluster, miR-92a, has been involved in lipid metabolism in hypoxic rats, which is relevant because hypoxia is known to induce lipolysis and inhibit fat synthesis [34]. Under these conditions, miR-92a expression levels decrease significantly in hypoxic rats compared with normoxic rats, suggesting that miR-92a deficiency could suppress lipolysis by regulating Fzd10/Wnt/β-catenin signaling [35]. Likewise, miR-19a-5p has recently been identified as a regulator of lipid metabolism in tilapia fish by participating in triglyceride synthesis, mainly because one of its target genes is the 3′UTR region of *DGAT2* (diacylglycerol O-acyltransferase 2) [36]. 

In addition to the role of the miR-17-92 cluster in lipid metabolism, a second related miRNA cluster has been also involved in metabolic pathophysiology. Deletion of the miR-106b-25 microRNA cluster (which contains miR-106b, miR-93, and miR-25) attenuates atherosclerosis in *Apoe* (apolipoprotein E) knockout mice, presumably by regulating plasma cholesterol levels [37]. Moreover, two of its three members, miR-106b and miR-93, were shown to impair cholesterol efflux [38,39]. Similarly, miRNAs from the miR-17 and miR-19 families are involved in lipid and cholesterol metabolism, which is mostly performed in subcellular organelles such as mitochondria, peroxisomes, and lysosomes. 

### 2.2. MiR-17-92 Family Members Participate in the Regulation of Autophagy

Besides enzyme-mediated lipid digestion and cholesterol metabolism, other metabolic lipid-related processes, such as vesicle trafficking and autophagy, are also essential for normal cell functioning. Vesicle trafficking is a means of intracellular transport; however, it is also implicated extracellularly, especially through exosomes. It is relevant to mention that miRNAs from the miR-17-92 cluster have been located within exosomes [40,41,42]. This suggests that this miRNA cluster is involved in cell-to-cell communication through exosomes, and therefore, in paracrine regulation from one cell to another. 

Autophagy is a basic and conserved cellular mechanism involved in the intracellular degradation of proteins and organelles, or pathogens, through the formation of autophagosomes [43], and it constitutes an extremely relevant metabolic process that needs to be highly regulated. As will be explained below, all members of the miR-17-92 cluster have been separately implicated in the regulation of autophagy in different situations. Paradoxically, there are virtually no studies involving all of them in the regulation of this central cellular phenomenon, probably because their role in autophagy has only recently been described.

The most-studied member of this cluster concerning autophagy is miR-17-5p [44]. Initially, Comincini et al. [45] identified miR-17-5p as a modulator of different autophagy-related proteins (ATGs), demonstrating that anti-miR-17-5p administration results in an increase in MAP1LC3B (microtubule-associated protein 1 light chain 3 beta) and ATG7 (autophagy-related 7) protein expression, and subsequently, yields an activation of autophagy through autophagosome formation in glioblastoma T98G cells. ATG7 is one of the master regulators of the autophagy process, controlling autophagosome formation and vesicle progression [46], and is a direct target of miR-17-5p. Also, it has been recently demonstrated that miR-17 is downregulated in chemoresistant non-small-cell lung cancer (NSCLC) cells. In these same cells, a lncRNA, *BLACAT1* (BLACAT1 overlapping LEMD1 locus), was shown to be significantly upregulated, together with *ATG7*, *ABCC1* (ATP binding cassette subfamily C member 1), LC3-I/II, and *BECN1* (beclin 1). Then, it was demonstrated that *BLACAT1* targets miR-17 and negatively regulates it, and thus promotes *ATG7* expression, which suggests that autophagy may be a novel target for overcoming drug resistance [47]. Additionally, two proteins relevant to autophagy and the autophagosome, ULK1 (Unc-51 like autophagy activating kinase 1) and LC3I/II, have been also identified as targets of miR-17-5p, being capable of down-regulating its expression in macrophage RAW264.7 cells in response to mycobacterial infection and therefore modulating phagosomal maturation [48]. 

Likewise, the levels of the anti-apoptotic myeloid cell leukemia 1 (*MCL1*), an apoptosis regulator member of the BCL2 family, and its transcriptional activator *STAT3* (signal transducer and activator of transcription 3) are down-regulated by miR-17-5p (*MCL1* suppresses autophagy through its ability to sequester *BECN1*), and overexpression of miR-17-5p also inhibits the phosphorylation of PRKCD (protein kinase C delta). Therefore, the miR-17-PRKCD-STAT3-MCL1 pathway emerges as a key regulating axis of autophagy during M. tuberculosis infection [49]. Further evidence extends these results to hepatic ischemia/reperfusion injury (IRI), in which high expression of miR-17-5p upregulates autophagy to promote hepatic IRI through the suppression of *STAT3* expression [50]. Similarly, miR-17-5p overexpression decreases the transcription of *STAT3* in vascular smooth muscle cells subjected to hypoxia-induced autophagy. Also, miR-17-5p could directly target *BECN1*, which mediates irradiation-induced autophagy activation in a glioma cell line [51]. Furthermore, miR-17 inhibition promotes cisplatin-induced autophagy of tongue squamous cell carcinoma CAL-27 cells through the *STAT3* pathway [52]. 

Moreover, miR-17-5p is also able to suppress autophagy in an osteoarthritis (OA) mice model, in which decreased miR-17-5p expression induces autophagy mainly through suppressing the expression of another of its targets, *SQSTM1*/*p62* (sequestosome 1), thereby contributing to osteoarthritis progression [53]. Interestingly, miR-17-5p has also been recently involved in osteosarcoma pathophysiology. It is upregulated in osteosarcoma cell lines and induces autophagy by targeting *PTEN* [54]. In addition, it has been recently demonstrated that miR-17-5p targets *Mfn2* (Mitofusin 2), a mitochondrial fusion protein that plays a role in balancing autophagy and inhibits its expression, activating the PI3K/AKT/mTOR pathway and suppressing autophagy to promote cardiac hypertrophy [55]. Again, miR-17-5p downregulation inhibits autophagy and myocardial remodeling after myocardial infarction by targeting the STAT3 pathway [56]. It is relevant to mention that inhibition of *MTOR* (mechanistic target of rapamycin kinase) by miR-17-5p upregulates autophagy and slows down the aging process [57]. 

Consequently, miR-17-5p may be a novel central miRNA regulator in stress-induced cellular mechanisms, considering stressful situations like pathogen infections, hypoxia, irradiation, inflammation, hypertrophy, or cancer chemotherapy and chemoresistance, reacting by mediating autophagy responses that in turn appear to be regulated by miR-17-5p. 

A decade ago, miR-20a was also first appointed as an autophagy-related gene because of its ability to negatively regulate autophagy in C2C12 myoblasts [58]. Since then, cumulative evidence has been shown to support this idea. Under hypoxia, *HIF1A* (hypoxia-inducible factor 1 subunit alpha) suppresses miR-20a, which negatively regulates *ATG16L1* (autophagy-related 16 like 1), an autophagy-related gene, solidifying the HIF1A-miRNA-20a-ATG16L1 regulatory axis as a critical mechanism for hypoxia-induced autophagy in osteoclast differentiation [59]. Additionally, in U-251 glioma cells, *EMAPII* (low-dose endothelial-monocyte-activating polypeptide-II) also downregulates miR-20a and induces autophagy by subsequently increasing the expression of *ATG5* (autophagy-related 5) and *ATG7*, both of which are miR-20a targets [60]. miR-20a also directly targets and inhibits *ATG7* and *TIMP2* (TIMP metallopeptidase inhibitor 2) in SiHa cells [61]. Depletion of miR-20a suppresses proliferation and autophagy and promoted apoptosis by increasing the expression of one of its targets, *THBS2* (thrombospondin 2), in cervical cancer cells [62]. Interestingly, resveratrol induces cell autophagy and decreases the inhibitory effect of miR-20a on another of its targets, *PTEN*, thus activating the PTEN/PI3K/AKT signaling pathway to attenuate liver fibrosis [63]. 

On the other hand, overexpression of miR-20a, induced by mycobacterial infection, inhibits the autophagy process in macrophages by targeting *ATG7* and *ATG16L1* and suppressing their expression [64]; miR-20a is also overexpressed in breast cancer, where it targets several genes related to autophagy, such as *BECN1*, *ATG16L1*, and SQSTM1/p62, downregulating them. Consequently, miR-20a inhibits autophagic flux and lysosomal proteolytic activity. If endogenous miR-20a is blocked, autophagic flux is increased [65]. miR-20a-5p gain-of-function also inhibits autophagy in ovary cancer via *DNMT3B* (DNA methyltransferase 3 beta)-mediated DNA methylation of *RBP1* (retinol-binding protein 1) [66]. Therefore, lowering miR-20a expression activates autophagy flux by upregulating the expression of autophagy-related proteins, and contrarily, overexpression of miR-20a inhibits autophagy and lysosomal proteolytic activity by downregulating them. 

The first member of the miR-17-92 cluster that was involved in autophagy regulation was miR-18a. Its ectopic overexpression in HCT116 colon cancer cells promotes autophagy by upregulating *ATM* (ataxia telangiectasia mutated) gene expression, a Ser/Thr protein kinase, and a member of the PI3K (phosphoinositide 3-kinase)-related protein kinase (PIKK) family, and by inhibiting mTORC1 (mechanistic target of rapamycin complex 1) activity [67]. Additionally, miR-18a-5p downregulates apoptosis and upregulates resveratrol-induced autophagy in the kidney podocytes of db/db mice (diabetic mice), also through targeting of the *Atm* gene [68]. Similarly, miR-18a induces the degradation of the oncogenic protein hnRNPA1 (heterogeneous nuclear ribonucleoprotein A1) by forming a complex that is then degraded through the autolysosomal pathway, enhancing the autophagy pathway itself [69]. Furthermore, miR-18a-5p promotes autophagy in NSCLC, enhancing not only autophagosome formation but autophagy flux [70]. Similarly, downregulation of miR-18a promotes autophagy by upregulating *BDNF* (brain-derived neurotrophic factor) expression and by inactivating the AKT/mTOR axis in hypoxia-induced rat cardiomyocytes, modeled through the H9c2 cell line. Likewise, the upregulation of *BDNF* suppresses cell senescence through the downregulation of miR-18a [71]. 

A less-studied microRNA of the miR-17-92 cluster concerning autophagy is miR-19a. Its overexpression in cardiomyocytes ameliorates hypoxia-induced cell death by switching from apoptosis to autophagy through downregulation of its specific target *BCL2L11* (BCL-2-like 11), an apoptotic activator [72]. In addition, reduced expression of miR-19a by propofol impedes autophagy and apoptosis caused by glutamate in PC12 cells, through an activation of AMPK (AMP-activated protein kinase) and mTOR signaling pathways [73].

Finally, and even less studied, miR-92a-3p has also been involved in endothelial cell autophagy and cardiomyocyte metabolism. In vitro, its inhibition promotes autophagy through the expression of *ATG4A* (autophagy-related 4 cysteine peptidase) [74]. Interestingly, the lncRNA *MALAT1*, acting as a sponge in cardiac cells induced to senescence, was identified as an exosomal transferred RNA that represses miR-92a-3p expression to unblock *ATG4A* [75]. 

Taken together, all this experimental evidence strongly suggests the direct involvement of the miR-17-92 cluster in the regulation of autophagy. To move forward, we must emphasize here that both autophagy and intracellular metabolism are functions mainly performed by lysosomes, and consequently, regulation of these functions must be connected, somehow, with the regulation of lysosomal functioning, and vice versa, as we will discuss below.

### 2.3. MALAT1, a Long Non-Coding RNA That Binds miRNAs of the miR-17-92 Cluster, Is Also Involved in the Regulation of Atherosclerosis and Autophagy

Many miRNAs are subject to various regulatory mechanisms, and one of them involves their inhibition through the binding of lncRNAs. These lncRNAs interact with miRNAs in a functional network that affects several processes. Only a few lncRNA are well-conserved among species, one of them being *MALAT1*, which blocks numerous miRNAs by providing non-functional binding sites and plays a crucial role in atherosclerosis [76,77,78] and autophagy [79,80]. Although to date there are no works that investigate the relationship between *MALAT1* and the miR-17-92 cluster, it is noteworthy that there are six works that correlate this lncRNA with two of the members of the cluster: miR-17-5p and miR-92a. 

Moving on to miR-17-5p, it is interesting to note that diabetic patients who smoke present higher serum levels of *MALAT1* and lower miR-17 levels in comparison to the serum of nonsmokers. This could be explained by the fact that cigarette smoke extract inhibits insulin production by upregulating *TXNIP* (thioredoxin interacting protein) via *MALAT1*-mediated downregulation of miR-17 [81]. Also, it has been reported that *MALAT1* expression increases in HeLa and CaSki cells treated with Cas-II-gly, together with a suppression of Wnt (Wingless-related integration site) signaling. Hence, *MALAT1* inhibits *FZD2* (frizzled class receptor 2) expression by targeting miR-17-5p via inactivation of the Wnt signaling pathway [82]. Finally, and more interesting for our reviewed topic, as we mentioned before, *MALAT1* regulates cholesterol accumulation via the microRNA-17-5p-*ABCA1* axis [24].

On the other hand, miR-92a can also be regulated by *MALAT1*. In human coronary artery endothelial cell (HCAEC)-derived exosomes under hyperbaric oxygen conditions, *MALAT1* is overexpressed, suppressing miR-92a expression, which in turn unblocks KLF2 (Kruppel like factor 2) to enhance angiogenesis [83]. These results have been replicated in a rat model of cardiac infarction [84]. Additionally, as previously commented, *MALAT1* sponges miR-92a-3p expression to inhibit cardiac senescence by targeting *ATG4A* [75]. 

Considering that *MALAT1* has been largely associated with both atherosclerosis and autophagy, that the miR-17-92 cluster is greatly involved in these two processes, and that its expression is directly regulated by *MALAT1*, analyzing this relationship in depth may yield interesting results. 

## 3. Lysosomes Are Fundamental Organelles for Cellular Metabolism and Autophagy

Cellular metabolism is highly compartmentalized within each cell, made possible by the endomembrane system. Although the mitochondrion is essential to produce energy from metabolic sources (both the Krebs cycle and the beta-oxidation of fatty acids take place here), other organelles, such as peroxisomes and lysosomes, are also fundamental for maintaining cellular metabolism. Besides cellular detoxification, peroxisomes also perform beta-oxidation of fatty acids, and lysosomes constitute the main intracellular digestive system of the cell. Consequently, it is reasonable to deduce that general pathways of metabolic regulation may affect the function of more than one organelle. Therefore, if the miR-17-92 regulatory cluster can perturb genes related to mitochondrial metabolic function, it could be also related, in some way, to genes involved in lysosomal metabolic function. 

Lysosomes are intracellular organelles that, in form of small vesicles, participate in several cellular functions, mainly digestion, but also vesicle trafficking, autophagy, nutrient sensing, cellular growth, signaling [85], and even enzyme secretion. The membrane-bound lysosome has long been regarded as the waste management and recycling facility of the cell because of its many hydrolytic enzymes used to digest various biomolecules. For these digestive enzymes to work properly, the pH inside the organelle must be highly acidic in comparison to the neutral cytoplasm [86]. More recently, the lysosome has been proven to be a dynamic key signaling hub involved in various pathways related to cellular adaptation, immunity, metabolism, and intracellular communication. Cells have between 50 and 1000 lysosomes in their cytoplasm, and the importance of these structures in cell homeostasis is highlighted by the severity of phenotypes present in LSDs [87].

### 3.1. Lysosomal Storage Diseases Affect Cellular Metabolism

LSDs are a heterogeneous group of conditions brought about by congenital inborn errors of metabolism (IEM), characterized by lysosomal dysfunction that led to the pathogenic accumulation of diverse molecules. Although the underlying basic mechanism behind the over 70 LSDs known to date is mostly the same, the accumulated or stored substance and the clinical manifestations vary widely among particular conditions [88]. Most LSD are classified as rare or ultra-rare conditions, but when put together, they have a joint prevalence of 1 in 5000 to 5500 live births [88,89] and belong to the even larger group of inborn errors of metabolism with a strong genetic component. These disorders are all monogenic and mostly have an autosomal recessive pattern of inheritance except for a few X-linked diseases, such as Fabry disease and Hunter syndrome [88]. As was mentioned before, although lysosomal dysfunction has a central role in all these diseases, the mechanisms that cause this to happen, as well as the pathways that lead to cell death, differ widely among them [90]. The specific organ damage involved in each of these conditions is different, but most of them seem to have a strong effect on the central nervous system, with many of them deemed neurodegenerative, which indicates a special susceptibility to lysosomal dysfunction in neurons and other nervous system cells [88,90]. 

Being mainly a genetic disease, epigenetic factors must also play relevant roles in LSD. Some of these epigenetic regulatory elements are miRNAs that function as regulatory molecules, which have been linked to several physiological processes as well as numerous diseases, among which, their role in LSD pathology is starting to be acknowledged [91,92]. In addition, miRNAs are recognized as novel regulators of cholesterol biosynthesis and metabolism [93], which are tightly controlled by the expression and proteolytic activation of the sterol regulatory element-binding proteins (SREBPs). An intricate network of miRNAs regulates gene expression of certain key genes, such as *APP* (amyloid-beta precursor protein), which appears to be regulated by multiple miRNAs, some of them being members of the miR-17 family (such as miR-20a-5p, miR-106a/b-5p) [93]. However, there is a lack of research on the role of ncRNA in lysosome functioning and particularly in LSD [92]. We must also consider, regarding miRNAs, that not only those produced within the cell are relevant to define the regulation of which they are capable, as exosomal miRNAs have an important role in cell signaling. Consequently, it is necessary to keep the possibility of the role of exosomes in LSD in mind [94]. 

### 3.2. MiRNAs of miR-17-92 Cluster Involved in LSD and NPC

Although miRNAs are potentially vital in the pathogenic mechanisms underlying LSDs, much information remains to be discovered. Very little work has focused on differentially expressed miRNAs in LSDs, and even less on Niemann–Picks type C (NPC) disease. 

The first correlation between LSD and miRNAs was most probably a paper, published in 2010, by Ozsait and colleagues, which identified variations in the concentration of several miRNAs involved in lipid metabolism and molecular transport in NPC fibroblasts. Although only a small percentage of the upregulated miRNAs were specifically related to cholesterol and lipid metabolism, they found some members of the miR-17-92 cluster (miR-19a, miR-19b, and a related one, miR-106b) to be among the most downregulated miRNAs [95]. All of them are linked to the lipid and partly glycosphingolipid metabolic processes, but miR-19a and miR-19b, in particular, are related to cholesterol transport and cholesterol-related metabolic processes; this, together with cellular cholesterol accumulation, constitutes one of the most prominent phenotypes in these cells [95].

More recently, Niculescu et al. evaluated the potential of miR-92a in the reversal of hyperlipidemia in hamsters. They discovered that miR-92a targets ABCG4 (ATP-binding cassette G4) and NPC1 (NPC intracellular cholesterol transporter 1) proteins, and that anti-miR-92a restored ABCG4, NPC1, and SOAT2 (sterol O-acyltransferase 2) expression. Interestingly, they proposed that in vivo inhibition of miR-92a could be a potential approach to correct lipid metabolism dysregulation and even atherosclerosis [96].

Some other cumulative evidence independently links miRNAs from the miR-17-92 cluster with different LSDs. In the case of Gaucher disease (GD), caused by deficiency of GBA1 (glucocerebrosidase 1) enzyme [97], it has been reported that miR-19a-5p is one of the three miRNAs that strongly down-regulate SCARB2 (scavenger receptor class B member 2) expression, which is an important membrane receptor involved in GBA1 availability [98]. Also, protein Lrrk2 (leucine-rich repeat kinase 2), involved in the mitochondrial dysfunction pathway in GD is regulated by miR-19b-3p [99]. 

Another LSD is Mucopolysaccharidosis type I (MPS I), caused by deficiency of IDUA (alpha-L-iduronidase), which leads to the ubiquitous accumulation of two glycosaminoglycans (GAGs), dermatan, and heparan sulfates [100,101]. Pereira et al. examined gene expression of Neu1 (neuraminidase 1) and Ctsa (cathepsin A), two components of the lysosomal multienzyme complex (LMC) in the cerebellum of MPS I mice and controls. They found that miRNAs from the miR-17 family (miR-17, miR-20a, miR-20b, miR-93, miR-106a, and miR-106b) were predicted to bind them, thus suggesting that this family of miRNAs might play a role in the regulation of lysosomal multienzyme complex (*LMC*) gene expression [102]. 

Furthermore, studies focused on Fabry disease also found miRNAs from the miR-17-92 family to be involved in its pathology. This is the case for miR19a-3p, involved in TGF (transforming growth factor)-beta signaling pathways, which is significantly down-regulated in Fabry disease male patients with enzyme replacement therapy (ERT) [103]. However, in the case of Fabry disease, many other miRNAs have been involved [104]. 

Other LSDs also present alterations in miRNAs from the miR-17-92 family. This is the case for GM2-Gangliosidosis deficiencies, such as Tay–Sachs and Sandho diseases, in which a panel of nine miRNAs that included miR-19a have been identified as highly downregulated [105]. 

Regarding probable interventions and the use of miRNA therapies to alleviate the consequences of LSD, it is of interest to analyze the few results obtained in this approach. First, one miRNA that has been described as linked to LSD and neurodegenerative disorders and is not directly related to the miR-17-92 cluster is miR-155, which is considered an inflammatory master regulator. Nonetheless, ablation of the pro-miR-155 does not mitigate neuroinflammation or neurodegeneration in a vertebrate model of GD [106] and does not affect the neuroinflammatory trajectory in an infantile neuronal ceroid lipofuscinosis (INCL) mouse model, also known as CLN1-disease, a devastating neurodegenerative LSD [107]. Contrastingly, in 2010, Gentner et al. demonstrated that therapy with transplanted hematopoietic stem and progenitor cells (HSPCs) is useful in the treatment of LSD. Interestingly, they reported that members of the miR-17-92 cluster (miR-19, miR-93a, and miR-17-5p) were highly expressed in hematopoietic stem and progenitor cells [108] which could indicate that those precursors exert a modulatory effect to improve the metabolism in affected organisms, compensating the damage.

### 3.3. Niemann–Picks Disease Type C and the Possible Involvement of miR-17-92 Cluster in It

Niemann–Picks type C (NPC) disease is an LSD characterized by the pathogenic accumulation of unesterified cholesterol and other lipids due to mutations in the genes coding for the intracellular cholesterol transport proteins NPC1 (95% of cases) or NPC2 (NPC intracellular cholesterol transporter 2, 5% of cases), that leads to a progressive neurovisceral condition [109]. The disease prevalence is currently estimated to be 0.95 per 1 million people in the United States, but the diagnostic difficulty of atypical cases might make the real prevalence higher [110]. The genetic and hereditary mechanisms behind this disease and other LSDs are still to be completely elucidated, as most cases are compound heterozygotes and notable phenotypical differences have been found among siblings and twins with identical genetic variations. Because of this and the unusual genetics behind NPC, the role of miRNAs and other epigenetic mechanisms in its pathogenesis is particularly interesting [111].

Under normal conditions, the protein products of *NPC1* and *NPC2* are thought to work in unison to export cholesterol and other molecules from the lysosome to other organelles, such as the mitochondria. NPC2 collects cholesterol from the lysosomal lumen and transports it to the lysosomal-membrane-bound NPC1, which then translocates it to the exterior [112]. It is relevant to note that, during neuronal aging, the activation of the AKT-mTOR pathway triggers the degradation of NPC1 protein, which induces the accumulation of cholesterol in endosomal compartments [113], similarly to what occurs in *NPC1* mutant and Niemann–Picks disease. Because the AKT/mTOR pathway can be upregulated, at least indirectly, by different members of the miR-17-92 cluster, such as miR17-5p, miR-20a, and miR18a (see previous mentions of this), it is logical to suppose that there is a strong relationship between this miRNA cluster and the metabolic consequences of Niemann–Picks disease.

## 4. Concluding Remarks

Although the role of the miR-17-92 cluster during embryonic development in cell differentiation, growth, and morphogenesis, as well as in oncogenesis, has been well-established, its role in cell metabolism, mainly in lipid and cholesterol flux under pathological conditions such as atherosclerosis and in autophagy as a cellular response to different situations is not established. Here, we present comprehensive up-to-date experimental evidence that supports the fundamental role of the miR-17-92 cluster in regulating cellular energetic metabolism, mainly lipid and cholesterol flux and as a key regulator in atherosclerosis, as well as a critical participant in regulating autophagy. Because these cellular functions are closely related to lysosomes, we also propose that the miR-17-92 cluster would be somehow involved in LSD effects. A summary of the molecular network in which the miR.17-92 cluster is involved is presented in Figure 1.

Alterations in the transport mechanism driven by NPC1 and NPC2 cause the abnormal storage of cholesterol and other macromolecules in lysosomes and late endosomes. This has many consequences, such as a general slowing down of the endocytic process, which prevents the binding of cholesterol vesicles to endosomes, disruption of autophagy, deregulation of proteins (due to lack of cholesterol in the endoplasmic reticulum and Golgi apparatus), and mitochondrial damage, which then leads to eventual cell death [112,114]. Accordingly, alterations that affect the lysosomal-mitochondria relationship and their metabolic equilibrium generate a defective metabolism, which contributes to disease progression [115]. Consequently, the identification of regulatory molecular links between these two organelles will most probably cause the rise of novel targets for the treatment of NPC. Therefore, we propose that members of the miRNA-17-92 cluster could be relevant actors in the clinical consequences of LSD, and therefore, could be considered pharmacological targets to, at least partially, alleviate this pathological condition.

## Figures and Tables

**Figure 1 cells-11-02991-f001:**
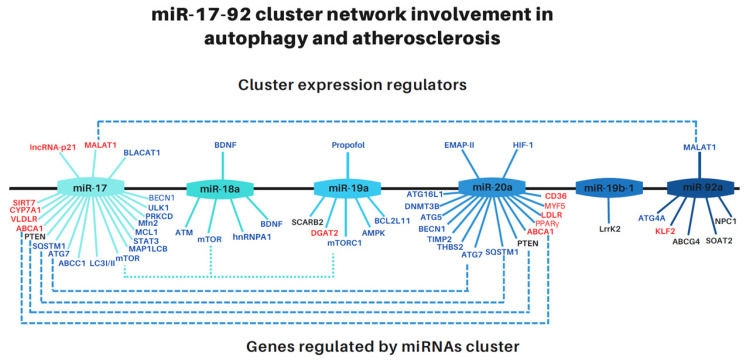
**Regulatory network of the miR-17-92 cluster involved in autophagy and atherosclerosis**. The cluster is composed of six miRNAs. The upper part of the line shows which molecules could regulate the expression of each miRNA. At the bottom of the line, the gene targets for each miRNA are illustrated. Those that match are linked by lines. Genes in bold are those verified experimentally and non-bold are hypothetical targets with a binding sequence for the miRNA. In red are genes involved in atherosclerosis, in black are genes involved in autophagy, and in blue are those involved in lysosomal storage diseases. Abbreviations: cancer-associated transcript 1 (BLACAT1), metastasis-associated lung adenocarcinoma transcript 1 (*MALAT1*), cytochrome P450 family 7 subfamily A member 1 (*CYP7A1*), protein light chain 3 (LC3-I to LC3-II), adenosine triphosphate (ATP)-binding cassette A1 (*ABCA1*), autophagy-related 7 (*ATG7*), p62/sequestosome 1 (*SQSTM1*), mitofusin 2 (*Mfn2*), sirtuin-7 (*SIRT7*), autophagy-activating kinase 1 (*ULK1*), diacylglycerol O-acyltransferase 2 (*DGAT2*), fatty acid translocase CD36 (*CD36*), low-density lipoprotein receptor (*LDLR*), autophagy-related 5 (*ATG5*), autophagy-related 16L1 (*ATG16L-1*), beclin-1 (*BECN1*), thrombospondin 2 (*THBS2*), phosphatase and tensin homolog (*PTEN*), DNA methyltransferase 3 beta (*DNMT3B*), leucine-rich repeat kinase 2 (*LRRK2*), ATP-binding cassette G4 (*ABCG4*), autophagy-related 4 (*ATG4*),Niemann-Pick C1 (*NPC1*), very low-density lipoprotein receptor (*VLDLR*), ATP binding cassette subfamily C member 1 (*ABCC1*), mTOR complex 1 (*mTORC1*), mechanistic target of rapamycin kinase (mTOR), microtubule-associated protein 1 light chain 3 beta (*MAP1LC3B*), signal transducer and activator of transcription 3 (*STAT3*), PRKCD protein kinase C delta (*PRKCD*), ataxia telangiectasia mutated (*ATM*), heterogeneous nuclear ribonucleoprotein A1, (*hnRNPA1*), brain-derived neurotrophic factor (*BDNF*), scavenger receptor class B member (*SCARB2*), TIMP metallopeptidase inhibitor 2 (*TIMP2*), Kruppel-like factor 2 (*KLF2*), sterol O-acyltransferase 2 (*SOAT2*), myogenic factor 5 (*MYF5*), proliferator-activated receptor-γ (*PPARγ*), anti-apoptotic myeloid cell leukemia 1 (*MCL1*), long intergenic non-coding RNA-p21 (*lincRNA-p21*).

## Data Availability

Not applicable.

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
