# Peer review of "The Role of the miR-17-92 Cluster in Autophagy and Atherosclerosis Supports Its Link to Lysosomal Storage Diseases"

_cells, 2022, doi:10.3390/cells11192991_

Round 1

Reviewer 1 Report

This manuscript compiled, analyzed, and discussed current evidence that highlights the cluster’s fundamental role in regulating cellular energetic metabolism (mainly lipid and cholesterol flux), its role as a key regulator of atherosclerosis, as well as a critical participant in autophagy regulation. This study also propose and provide experimental data to support that the miR-17-92 cluster could be involved in lysosomal storage diseases (LSD) pathogenesis and effects. This study is potential interesting for relevant studies, but some contents should be carefully checked and revised. The current version is not enough to present the whole study.

1.   There are many miRNA gene clusters and families in miRNA world. It is better to introduce why authors select the miR-17-92 cluster, and relevant homologous miRNAs may be also crucial that should be discussed in the manuscript.

2.   The authors say that they provide experimental data to support that the miR-17-92 cluster could be involved in lysosomal storage diseases (LSD) pathogenesis and effects. But these experimental data were only derived from published literatures. The original description is prone to ambiguous.

3.   The manuscript should contain more intuitive Figures or tables to present the relevant contents or results.

4.   Whether some circRNAs can act as miRNA sponge to contribute to autophagy and atherosclerosis?

5.   Although this study mainly reported the role of miR-17-92 cluster in autophagy and atherosclerosis, the whole contents only focus on target mRNAs of several miRNAs. It is better to show the target mRNA landscapes of these cluster miRNAs, and then integrate the potential expression and functional relationships to discuss their biological roles in autophagy and atherosclerosis.

Author Response

We are deeply grateful to the reviewer for his positive comments on our manuscript, and for the time dedicated to elaborating these appreciated and valuable comments. We hope that our point-by-point explanations will satisfy the reviewer's request, and we are willing to consider further comments and suggestions to improve our work.

1- There are many miRNA gene clusters and families in miRNA world. It is better to introduce why authors select the miR-17-92 cluster, and relevant homologous miRNAs may be also crucial that should be discussed in the manuscript.

ANSWER:  We are studying autophagy processes, particularly in lysosomal storage diseases (LSD), so when we performed a review on the miRNAs described as participanting in this process, the miRNA cluster miR-17-92 caught our attention. We realized that although this cluster has been described as relevant for various types of cancers (lines 56-58) and for vascular integrity and angiogenesis (lines 58-59), it also has a relevant involvement in lipid metabolism, atherosclerosis and autophagy, which has not been sufficiently recognized. That is why we selected this cluster of miRNAs to review the current information in the literature in this regard and organize and summarize it to support our proposal. Therefore, this review was not intended to cover the totality of miRNAs that might be involved in the chosen aspects of autophagy, lipid metabolism and atherosclerosis. Rather, it is intended to support the suggestion that the miR-17-92 cluster is directly involved in these processes and, consequently, in LSD (explained in lines 27-29).

2- The authors say that they provide experimental data to support that the miR-17-92 cluster could be involved in lysosomal storage diseases (LSD) pathogenesis and effects. But these experimental data were only derived from published literatures. The original description is prone to ambiguous.

ANSWER: We thank the reviewer for noticing this aspect. To avoid ambiguity, we rewrote the sentence as follows: “we also provide experimental data from the literature to support our proposal that the miR-17-92 cluster could be involved in the pathogenesis and effects of lysosomal storage diseases (LSD).”

3- The manuscript should contain more intuitive Figures or tables to present the relevant contents or results.

ANSWER: We understand the point raised by the reviewer. In this review we decided to summarize the current findings of the miR-17-92 cluster in the regulation of cellular lipid metabolism, focusing on its role in autophagy, by carefully organizing the information in subheadings, consequently, the structure was chosen in this way. Thus, by following the subheadings, readers will be able to easily find the corresponding information. For now, there is only one figure illustrating the network of target genes of the mir-17-92 cluster experimentally involved in autophagy and atherosclerosis (which is the main objective of this review). In this figure, readers will find all the target genes so far described (which are supported by experimental data) that this cluster of miRNAs regulates, and those that regulate its members. Additionally, this figure clearly shows which of those targets are regulated by more than one member of the cluster. From this figure it is clear which miRNAs are more studied, and which are less studied, in each of the two biological processes referred to, showing a parallelism and thus suggesting possible options for further study. Nevertheless, if the reviewer considers that another figure, or a table, could be presented, we will be happy to elaborate them, based on a more specific suggestion from the reviewer.

4- Whether some circRNAs can act as miRNA sponge to contribute to autophagy and atherosclerosis?

ANSWER: Yes, this is precisely why we included in this review a specific subheading on MALAT1 (lines 271-301), which is involved in autophagy and atherosclerosis by acting as a sponge for cluster 17-92 miRNAs. Although other lncRNAs can perform this function, so far, this is the only one described that act on this cluster of miRNAs and in these two processes.

5- Although this study mainly reported the role of miR-17-92 cluster in autophagy and atherosclerosis, the whole contents only focus on target mRNAs of several miRNAs. It is better to show the target mRNA landscapes of these cluster miRNAs, and then integrate the potential expression and functional relationships to discuss their biological roles in autophagy and atherosclerosis.

ANSWER: We thank the reviewer for the opportunity to clarify this aspect. We agree with the reviewer that an alternative approach to analyze the role of a group of miRNAs in some specific cell function is by considering all their hypothetical gene targets, and perform an in silico analysis, however to support the information in this review, we selected a different approach by focusing only in those target genes that have been experimentally demonstrated as targets for the miRNAs in the considered cluster, and that have been experimentally connected precisely with the cellular functions considered, mainly autophagy.

Reviewer 2 Report

In this interesting review manuscript, authors summarized the current findings of miR-17-92 cluster in the regulation of cellular lipid metabolism, with focusing on their role in autophagy.  Authors also highlighted potential implication of disrupted miR-17-92 pathway in pathogenesis of lysosomal storage diseases.  In general, the manuscript was nicely organized, and almost all updated information regarding miR-17-92 studies was collected and logically presented. 

However, evidence supporting the role of miR-17-92 cluster in lipid metabolism and autophagy is limited and premature in general.  All results collected in the current manuscript seem associated nature but not causal relationship.  These available findings only suggest potential role of miR-17-92 cluster in the regulation of cellular lipid metabolism.  Although individual member of miR-17-92 family modulates lipid metabolism, whether miR-17-92 cluster functions together remains largely unexplored yet and should be clarified in the manuscript.  Most studies referred here just show an association of changes in individual member of miR-17-92 family with cellular metabolism and atherosclerosis but do not show they function together as a group. 

In addition, studies investigating miR-17-92 in pathogenesis of lysosomal storage disease are limited and only provide a litter significant information.

Minor concern:

  1. Abbreviation should appear at the first time and just once.
  2. “Lysosomal” should not be capitalized in the title.       

Author Response

We thank the reviewer for his/her positive comments on our manuscript, on the interestingness of the manuscript, as well as on its organization and presentation of the most up-to-date information regarding the miR-17-92 cluster.

Although the reviewer's comments are in a couple of paragraphs, we subdivide the first paragraph to explain the different aspects commented by the reviewer, as follows:

1 - However, evidence supporting the role of miR-17-92 cluster in lipid metabolism and autophagy is limited and premature in general. All results collected in the current manuscript seem associated nature but not causal relationship. These available findings only suggest potential role of miR-17-92 cluster in the regulation of cellular lipid metabolism.  Although individual member of miR-17-92 family modulates lipid metabolism, whether miR-17-92 cluster functions together remains largely unexplored yet and should be clarified in the manuscript.

ANSWER: We thank the reviewer for emphasize this aspect. This is a propositional review, in which we argue that the miR-17-92 cluster could be highly relevant to autophagy and to lipid metabolism and atherosclerosis. Consequently, in agreement with the reviewer’s request, we have added lines 77 to 80 as follows: “However, the findings available so far, only suggest a potential role of miR-17-92 cluster in the regulation of cellular lipid metabolism, then, although individual members of the miR-17-92 family modulates lipid metabolism, whether the miR-17-92 cluster functions as a whole remains largely unexplored.”

2 - Most studies referred here just show an association of changes in individual member of miR-17-92 family with cellular metabolism and atherosclerosis but do not show they function together as a group.

ANSWER: We agree with the point raised by the reviewer that the available information is still limited, however, as we present in this review, it exists and is consistent with the possibility we propose that, just as a has been extensively studied during development and in cancer, this cluster also acts together in autophagy and atherosclerosis. We also agree that the information collected is suggestive of a potential role that warrants further exploration, in this regard we here organize the available experimental information that supports the potential role of this cluster, which may serve as a basis for future research.

Our arguments are set out throughout the review as follows:

Relevance of the miR-17-92 cluster (lines 44-49)

Implicated as a key factor in metabolic cellular reprogramming (lines 63 to 69)

Possibly involved in the regulation of lipid metabolism (lines 70-77)

Relationship between lipid metabolism, atherosclerosis and lysosomal disfunction and possible involvement of the miR-17-92 cluster (lines 84-90)

The following subheadings on the key regulatory function of this miRNA cluster in atherosclerosis (lines 91-141), and autophagy (lines 142-264). Concluding in lines 265-266.

The last part of the review contains the lysosomes nexus, which is related to lipid metabolism, autophagy regulation, and atherosclerotic dysfunction. Presenting the evidence reported in the literature on the involvement of miRNas of the miR-17-92 cluster in these aspects.

Accordingly, in this review we compile, analyze, and discuss the current evidence highlighting  that the cluster could have a fundamental role in regulating cellular energetic metabolism (mainly lipid and cholesterol eflux), in addition to the role of all its members as key regulators of atherosclerosis, as well as critical participants in the regulation of autophagy.

Therefore, to the best of our knowledge, this proposal is novel and presents a solid basis to support further studies on the involvement of this cluster of miRNAs in autophagy, as well as in atherosclerosis and in relation to lysosomal storage diseases.

Specifically, this aspect raised by the reviewer is commented on lines 157-161 as follows:

As will be explained below, all members of the miR-17-92 cluster have been separately implicated in the regulation of autophagy in different situations. Paradoxically, there are virtually no studies involving all of them in the regulation of this central cellular phenomenon, probably because their role in autophagy has only recently been described.”

And in lines 265-266 as follows: “Taken together, all this experimental evidence strongly suggests the direct involvement of the miR-17-92 cluster, as a whole, in the regulation of autophagy”.

3 - In addition, studies investigating miR-17-92 in pathogenesis of lysosomal storage disease are limited and only provide a litter significant information.

ANSWER: We thank the reviewer for the opportunity to comment on this aspect. This is precisely one of the key points of this review article, to emphasize the relevance of this cluster and to highlight the importance of doing more research on it.

This aspect raised by the reviewer is commented on lines 358-361 as follows: “Although miRNAs are potentially vital in the pathogenic mechanisms underlying LSDs, much information remains to be discovered. Very little work has focused on differentially expressed miRNAs in LSDs, and even less on Niemann-Picks type C (NPC) disease.”

Minor concern:

Abbreviation should appear at the first time and just once.

ANSWER: we revised again this aspect

“Lysosomal” should not be capitalized in the title.

ANSWER: was corrected

Round 2

Reviewer 1 Report

The manuscript has been carefully revised and improved. Thanks.

Reviewer 2 Report

None